# Low-Affinity/High-Selectivity Dopamine Transport Inhibition Sufficient to Rescue Cognitive Functions in the Aging Rat

**DOI:** 10.3390/biom13030467

**Published:** 2023-03-03

**Authors:** Jana Lubec, Ahmed M. Hussein, Predrag Kalaba, Daniel Daba Feyissa, Edgar Arias-Sandoval, Anita Cybulska-Klosowicz, Mekite Bezu, Tamara Stojanovic, Volker Korz, Jovana Malikovic, Nilima Y. Aher, Martin Zehl, Vladimir Dragacevic, Johann Jakob Leban, Claudia Sagheddu, Judith Wackerlig, Marco Pistis, Merce Correa, Thierry Langer, Ernst Urban, Harald Höger, Gert Lubec

**Affiliations:** 1Programme for Proteomics, Paracelsus Medical University, 5020 Salzburg, Austria; 2Department of Pharmaceutical Sciences, Division of Pharmaceutical Chemistry, Faculty of Life Sciences, University of Vienna, 1090 Vienna, Austria; 3Department of Zoology, Faculty of Science, Al-Azhar University, Assiut 71524, Egypt; 4Department of Psychobiology, Universitat Jaume I, 12006 Castelló, Spain; 5Neurobiology of Emotions Laboratory, Nencki Institute of Experimental Biology, 02093 Warsaw, Poland; 6Department of Analytical Chemistry, Faculty of Chemistry, University of Vienna, 1090 Vienna, Austria; 7Department of Biomedical Sciences, Division of Neuroscience and Clinical Pharmacology, University of Cagliari, 09042 Monserrato, Italy; 8Section of Cagliari, Neuroscience Institute, National Research Council of Italy (CNR), 09042 Cagliari, Italy; 9Department of Psychological Sciences, Behavioral Neuroscience Division, University of Connecticut, Storrs, CT 06269, USA; 10Core Unit of Biomedical Research, Division of Laboratory Animal Science and Genetics, Medical University of Vienna, 2325 Himberg, Austria

**Keywords:** dopamine, DAT, learning and memory, aging, reward

## Abstract

The worldwide increase in cognitive decline, both in aging and with psychiatric disorders, warrants a search for pharmacological treatment. Although dopaminergic treatment approaches represent a major step forward, current dopamine transporter (DAT) inhibitors are not sufficiently specific as they also target other transporters and receptors, thus showing unwanted side effects. Herein, we describe an enantiomerically pure, highly specific DAT inhibitor, S-CE-123, synthetized in our laboratory. Following binding studies to DAT, NET and SERT, GPCR and kinome screening, pharmacokinetics and a basic neurotoxic screen, S-CE-123 was tested for its potential to enhance and/or rescue cognitive functions in young and in aged rats in the non-invasive reward-motivated paradigm of a hole-board test for spatial learning. In addition, an open field study with young rats was carried out. We demonstrated that S-CE-123 is a low-affinity but highly selective dopamine reuptake inhibitor with good bioavailability. S-CE-123 did not induce hyperlocomotion or anxiogenic or stereotypic behaviour in young rats. Our compound improved the performance of aged but not young rats in a reward-motivated task. The well-described impairment of the dopaminergic system in aging may underlie the age-specific effect. We propose S-CE-123 as a possible candidate for developing a tentative therapeutic strategy for age-related cognitive decline and cognitive dysfunction in psychiatric disorders.

## 1. Introduction

Cognitive decline during aging is a serious problem, especially in industrial societies with increasing life expectancy. Aging is accompanied by cognitive decline and is the main risk factor for most neurodegenerative disorders, which significantly affect the quality of life of individuals, their careers and the global economy. There is an urgent need for the development of therapeutic strategies to counteract aging-associated neuropathologies and slow down or reduce age-related cognitive decline.

Dopamine plays a fundamental role in learning and memory involving intrinsic neuronal circuits in the prefrontal cortex (PFC) and the hippocampus, as well as the neuronal network activity between these structures [1]. Therefore, treatments of various cognitive disorders in different psychiatric diseases using dopamine-targeting interventions have been undertaken [2,3]. The effects of dopamine as a target for treating cognitive disorders, however, depend on the memory systems engaged [4]. Spatial memories that structurally involve the hippocampus and PFC and declarative [5] and novelty [6] memory processes strongly depend on the neuromodulatory functions of dopamine. Further, the connectivity between these brain structures [7] as well as synaptic glutamatergic transmission within these structures is strongly influenced by dopamine mainly by the release from the ventral tegmental area (VTA). Basal ganglia can be involved in the modulation of non-declarative memories [8], such as the nucleus accumbens [9,10] and striatum [11] in conjunction with declarative memory. Age-related decline in cognitive abilities goes along with a degenerating dopaminergic system in various brain regions, including the hippocampus and PFC [12,13]. Pre- and post-synaptic dopaminergic components are thereby differently changed across the life-span, impairing cognitive functions such as working and episodic memory and decision making differently [14] in various age classes. Accordingly, growing evidence shows dopamine system impairment in Alzheimer’s disease [15,16], further contributing to the full-blown pathology [17,18].

The high variability in age-related changes in the dopaminergic system on one hand and the learning-task-dependent requirement of different dopamine receptors on the other hand make it difficult to target specific dopamine receptors to treat cognitive decline in general as well as age-dependently. Moreover, functional changes in dopamine via heteroreceptor complexes have been reported in the context of cognition and psychiatric diseases [19,20,21]. Targeting of the dopamine transporter (DAT) may circumvent the problems of task- and age-dependent functional differences of the dopaminergic system. DAT reuptakes extracellular dopamine into the cell, thus regulating the availability of the ligand for all membrane dopamine receptors. Accordingly, the inhibition of DAT leads to an increase in extracellular dopamine concentrations, from which a variety of dopamine receptors can benefit. It has been shown that DAT inhibitors affect hippocampal synaptic plasticity in the dentate gyrus [22,23] and CA1 [24], as well as working memory and long-term memory in different learning tasks (including spatial tasks) in rodents [22,25,26] and humans [27]. However, commercially available DAT inhibitors do not show sufficient specificity for DAT and also target NET and SERT to a certain extent. 

S-CE-123 (S enantiomer of CE-123), one of the leading modafinil analogues synthetized in our laboratory, possesses exceptionally high specificity for DAT. Even if S-CE-123 is only a weak DAT inhibitor, it clearly shows biological activity. A memory-enhancing effect has been shown in a spatial hole-board task in compromised young rats [28]. Camats-Perna et al. demonstrated that administration of CE-123 (racemate and S enantiomer) protects the consolidation of long-term social memory against interference for defined durations after learning [29]. Furthermore, an effect on cognitive flexibility [30] and a motivation effect in young [31] and old rats [32] have been recently reported. Importantly, we previously proposed reduced abuse liability of CE-123 [33]. 

This study aims to comprehensively evaluate the potency of a low-affinity DAT inhibitor in mitigating the cognitive decline in aging. 

## 2. Materials and Methods

### 2.1. Chemistry

(S)-5-((Benzhydrylsulfinyl)Methyl)Thiazole (S-CE-123) was synthesized according to previously described procedures [31] in Prof. Gert Lubec laboratory.

### 2.2. hDAT, hSERT and hNET Binding Assay

Binding affinity to the human dopamine transporter (hDAT), serotonin transporter (hSERT) and norepinephrine transporter (hNET) expressed in CHO cells was evaluated in in vitro radioligand binding assays. Specific binding was determined using [^3^H]BTCP, [^3^H]imipramine and [^3^H]nisoxetine for hDAT, hSERT and hNET, respectively. The IC50 values were calculated by non-linear regression analysis of the competition curves generated from mean replicate values. The K_i_ values were calculated using the Cheng–Prusoff equation (K_i_ = IC_50_/(1 + (L + K_D_))). Binding assays were carried out by Eurofins (Cerep SA, Celle l’Evescault, France) according to standardized and validated protocols.

### 2.3. Functional GPCRs Agonist and Antagonist Screening 

Large-scale functional GPCR screening for off-target activity was conducted by Eurofins Cerep (Celle l’Evescault, France) according to the standard and validated protocol defined by the contractor. A GPCR Assay Panel included 168 targets from 60 GPCR families (Appendix A). The final concentration of S-CE-123 in the assay was 2.8E-06 M. 

### 2.4. Kinase Binding Assays

A panel of binding assays covering 450 human kinases and disease-relevant mutations was performed by DiscoverX (KINOMEscan^®^, Freemont, CA, USA) according to the standard and validated protocol defined by the contractor. The final concentration of S-CE-123 used in the assay was 2.8E-06 M.

### 2.5. Determination of Plasma and Brain Levels of S-CE-123

The bioavailability of S-CE-123 after intraperitoneal (i.p.) administration in male Sprague Dawley rats (250–296 g) was evaluated by Evotec (Toulouse CEDEX, France). S-CE-123 at 10 mg/kg body weight was administered by i.p. route and blood samples and brains were collected at 0.25, 0.5, 1, 3, and 7 h post administration (3 rats/time point). No adverse reactions were observed after i.p. administration of S-CE-123. Plasma and brain levels of S-CE-123 were measured by LC-MS by a standard and validated protocol defined by the contractor, and the pharmacokinetic parameters were calculated from individual plasma concentrations by Phoenix WinNonLin7.0 (Certara, Princeton, NJ, USA), a non-compartmental model (see Appendix A). 

### 2.6. Neuronal Outgrowth

Cryopreserved rat primary cortical neurons (QBM Cell Science, Ottawa, ON, Canada) were quickly thawed and slowly diluted with neurobasal medium supplemented with L-glutamine, penicillin-streptomycin and B27 (NB/B27 medium) (Invitrogen, Carlsbad, CA, USA). After gentle centrifugation, neurons were resuspended with NB/B27 medium at a density of 4.0 × 10^6^ cells/mL. One hour prior to treatment, neurons were plated in 384-well plates pre-coated with laminin at a density of 10,000 cells per well. After treatment with S-CE-123 (0.2-100E-06 M) or vehicle, the neurons were maintained in a humidified environment at 37 °C with 5% CO_2_ for 72 h. Subsequently, the neurons were fixed for one hour with a 3.7% formaldehyde solution, washed with phosphate-buffered saline (PBS) and permeabilized for ten minutes with 0.5% Triton X-100. After another washing step, neurons were incubated for one hour with Hoechst 33342 (ThermoFisher Scientific, Waltham, MA, USA) and Anti-Beta III Tubulin antibody conjugated with AlexaFluor488 (AB15708A4, Millipore, Billerica, MA, USA). The neurons were evaluated for neurite outgrowth and cell health using the high-content imaging platform, ArrayScan VTi (Thermo Scientific). Valid neuron count, mean neurite average length and neurite total length per neuron were determined and reported. Raw data were normalized to the vehicle control and reported as a normalized % response. Dose–response graphs were generated with GraphPad Prism using a sigmoidal dose–response (variable slope) algorithm.

### 2.7. Animals and Animal Ethics

Rats used for behavioral experiments were male Sprague Dawley rats bred and maintained in the Core Unit of Biomedical Research, Division of Laboratory Animal Science and Genetics, Medical University of Vienna. Animals were housed in groups of two or three at 22 ± 2 °C, humidity: 55 ± 5%, on a 12 h light–dark cycle: light on at 7:00 am. The exact age and number of animals are described in the section for the specific behavioral test. All experiments on living animals were carried out in accordance with the 2010/63/EU guidelines and were approved by the Federal Ministry of Education, Science and Culture, Austria (BMBWF-66.009/235/V/3b/2018). All efforts were directed toward minimizing the number of animals used and their suffering.

### 2.8. Hole-Board Spatial Learning and Memory Task

A hole-board test was performed as previously described [34]. Young and old rats were on a calorie restriction diet (R/M Ered II ssniff^®^, Soest, Germany) for five weeks for young and four months for aged rats prior to the task after they had been fed standard rat chow (R/M-H ssniff^®^, Soest, Germany). Twenty-four hours before the start of the experimental procedure, the animals were transferred to a separate experimental room and kept there throughout the experiment individually in standard Makrolon cages filled with autoclaved woodchips. All behavioral trainings and tests were performed during the light phase of the light–dark cycle. 

Controlled food restriction was conducted to reduce the weight of the rats to 85% of their initial body weight. Apparatus was a black plastic board (1 m ×1 m) with sixteen regularly distributed holes (7 cm in diameter and depth) surrounded by transparent walls (30 cm high) enriched with proximal cues. Distal visual cues were located on the room walls. A second board below the testing arena was covered with food pellets to avoid olfactory orientation.

The experiment started with handling (10 min per day for four consecutive days) to acclimatize the animals to human interaction. In the following two days, animals were habituated to the hole-board apparatus (free exploration of the maze for 15 min each day with access to food pellets (dustless precision pellets, 45 mg, Bioserv, Flemington, NJ, USA)). The training was conducted over three consecutive days (five trials on day 1, four trials on day 2 and one trial on day 3) with an intertrial interval of 20 min. Four out of sixteen regularly arranged holes were baited. The pattern of baited holes remained the same during the entire test. Each trial lasted 2 min or until all four pellets were found. The board was cleaned with 1% Incidin^®^ after each trial in order to remove any olfactory cues. The hole visits and removals of pellets were noted for each trial. The Reference Memory Index (RMI) was calculated using the formula (first + revisits of baited holes)/total visits of all holes and was normalized to the number of first visits to baited holes. All rats visited more than forty holes in total over the ten trials. 

Forty naïve young (5 months old) rats and thirty-eight aged (22–23 months old) behaviorally characterized rats were used for treatment. S-CE-123 was freshly dissolved in 30% Kolliphor EL (BASF Pharma, Lampertheim, Germany; vehicle) and administered via i.p. every day 30 min before the start of the behavioral experiment throughout the training sessions. Rats received 1 mL/kg vehicle or drug administered at doses of 1, 5 and 10 mg/kg of body weight. Drug treatment did not impair general health conditions such as movement, gait, salivation, sedation, tremor, convulsion, diarrhea, etc. 

### 2.9. Morris Water Maze (MWM)

A second independent sample of twenty aged rats (21–22 months old) was tested in the MWM task. The water maze consisted of a circular black water tank (150 cm in diameter, 60 cm in height) filled with water (25 ± 1 °C) up to 40 cm. A black platform (10 cm × 10 cm) 1.5 cm below the water surface allowed the animals to escape from the water by climbing on it. The animals were handled two days prior to the experiment, followed by a habituation day during which the animals were allowed to explore the water maze for 2 min without platform. The training consisted of 4 days with 5 trials each. The trial duration was two minutes, and the intertrial interval was 20 min. If the animal did not find the platform, they were guided by hand to the platform in the southwest quadrant, which they climbed and remained on for 30 s. Thereafter, the rats were dried with a towel and set back into their keeping cages. On day 5, a probe trial of 30 s with a removed platform was performed. From day 6 to 9, a reversal training was performed with the same protocol as during the initial training but with the platform located in the opposite quadrant (northeast). On day 10 again, a 30 s probe trial without a platform was conducted. The latencies to escape from the water, the mean velocity and the travelled distances were calculated by a video tracking system (TIBE V.1.0; Vienna, Austria). The percentage of time spent in the four quadrants of the maze was calculated during the probe trials. 

Furthermore, 10 mg/kg S-CE-123 or vehicle (1 mL/kg of body weight) was administered via i.p. every day 30 min prior to the start of the behavioral experiment. 

### 2.10. Open Field (OF)

Twenty-five naive young (10 weeks) rats were used in this experiment. Animals were habituated to the experimental room twenty-four hours prior to the start of the experiment. The open field consisted of a black wooden board (1.20 m × 1.20 m) surrounded by white wooden walls (50 cm in height); light was adjusted to 150 lux. Each animal was placed in the middle of the open field arena and behavior was recorded for 10 min. Data analysis was performed using the AnyMaze software (Stoelting Co., Wood Dale, IL, USA) or TIBE software (TIBE V.1.0; Vienna, Austria). The novelty test consisted of one test session; 30 min prior to the test, each animal received 10 mg/kg S-CE-123 or vehicle (i.p., 1 mL/kg). The familiar test consisted of one habituation session on the first day, when all animals received the vehicle (30% Kolliphor EL, i.p., 1 mL/kg) and one test session on the following day (i.p., S-CE-123 or vehicle). In the familiar OF test, each animal received S-CE-123 at 10, 20, 0 and 1 mg/kg body weight with a wash-out period of nine days between each treatment.

### 2.11. Immunohistochemistry

One hour after the last trial in the hole-board task, rats were deeply anesthetized and perfused intracardially with ice-cold 4% paraformaldehyde (PFA) in 0.1 M PB (pH 7.4). Brains were removed, postfixed overnight in 4% PFA at 4 °C and cryoprotected in 30% sucrose solution for 48 h. Coronal sections (15 µm) were cut with a cryostat (Leica, Heidelberg, Germany), mounted on Superfrost Plus glass slides and stored at −20 °C until staining.

Frozen sections were incubated with 1% hydrogen peroxide for 10 min at 22 °C to quench endogenous peroxidase activity. Antigen retrieval was performed using 10 mM sodium citrate target retrieval solution pH6.0 (S236984, Dako, Agilent, Santa Clara, CA, USA) at 95 °C for 10 min, and sections were subsequently blocked in buffer containing 2% BSA, 5% normal goat serum (NGS) and 0.1% Triton X-100 in 0.1 M phosphate buffer (PB; pH 7.4) for 2 h at 22 °C. Slides were washed 3 times with 0.1 M PB for 10 min each, followed by incubation with primary anti-CD68 antibody (1:1000, MCA341GA, BIO-RAD, Hercules, CA, USA) in 0.1 M PB containing 1% NGS, 0.1% BSA and 0.1% Triton X-100 at 4 °C for 72 h. After three washing steps, sections were incubated for 2 h at 22 °C with biotinylated goat secondary antibody (1:200, BA-9200, Vector Laboratories Inc., Mowry Ave, CA, USA) and subsequently incubated with the avidin–biotin–peroxidase complex (PK-6100; ABC kit; Vector Laboratories Inc.). The peroxidase reaction was visualized by incubating the sections for 5 min with an ImmPACT DAB substrate (SK-4105; ABC kit; Vector Laboratories Inc.). Images were captured with a high-resolution digital slide scanner NanoZoomer 2.0 HT (Hamamatsu Photonics, Hamamatsu, Japan). Single images were exported from digitalized slides using the NDP.view2 software (Hamamatsu Photonics, Japan) containing dorsal hippocampal subregions. Quantitative analysis of immunoreactivity was performed in ImageJ (NIH, 1.50 g). The area of immunopositive structures was calculated for all exported images. No staining was detected in the absence of primary or secondary antibodies.

### 2.12. Statistical Analysis

No statistical method was used to predetermine sample size. The sample size was chosen based on previous studies in our laboratory. For animal experiments, animals were grouped in a random manner to reduce bias. Data were tested for normal distribution (D’Agostino-Pierson test) and equality of variances (F test for *t*-test and Bartlett test for ANOVAs). Behavioral data were analyzed using a one or two factor analysis of variance (ANOVA), with repeated measures when indicated followed by the Dunnett’s post hoc test. Differences between means were considered significant at the *p* < 0.05. All data are expressed as mean ± SEM and were analyzed using GraphPad Prism 7 software (GraphPad Software, San Diego, CA, USA). Detailed statistical parameters for specific experiments are described in the appropriate section or figure legends.

## 3. Results

### 3.1. Binding of S-CE-123 to Human Monoamine Transporters 

CHO cells expressing hDAT, hSERT or hNET were used to assess binding affinity with K_i_ value of 6.1E-07 M at hDAT, whereas there was no binding to SERT or NET (at concentration lower then 1E-04 M) (Figure 1A). S-CE-123 displayed a high selectivity for DAT over NET and SERT. Results are in line with reuptake data, showing that S-CE-123 is a relatively weak but very selective dopamine reuptake inhibitor at hDAT with IC50 = 2.8E-06 M [30]. 

### 3.2. Analysis of Plasma and Brain Concentrations of S-CE-123

Pharmacokinetic parameters were determined after intraperitoneal (i.p.) administration of S-CE-123 at 10 mg/kg body weight. The mean plasma S-CE-123 concentrations peaked within 15 min after i.p. administration (Cmax = 5649 ± 2692 ng/mL) and exhibited a short apparent terminal half-life (0.74 ± 0.21 h). Mean brain S-CE-123 concentrations reached a maximum of 4781 ± 919 ng/g at 15 min after i.p. administration, and the mean brain to plasma concentration ratio value was 0.96 ± 0.06. The estimated brain to plasma concentration ratio of >0.9 indicates that S-CE-123 is able to distribute across the blood–brain barrier. The plasma and brain concentrations of S-CE-123 at different time points are illustrated in Figure 1B,C. The pharmacokinetic parameters of i.p. administration of S-CE-123 in a rat show that the drug is rapidly absorbed, distributed and eliminated (see Appendix A). 

### 3.3. GPCRs and Kinome Screening 

In order to address the potential off-target activity, S-CE-123 was subjected to a large-scale screening for GPCR and kinase targeting. S-CE-123 at 2.8E-06 M (IC50 for DAT re-uptake inhibition) showed no agonist or antagonist activity (inhibition or stimulation < 40%) against any of the 168 GPCRs (see Appendix A) tested in the functional cellular assay. Similarly, in in vitro competition binding assay against 450 protein and lipid kinases (see Appendix A) representative of all major human kinase families, no positive hit was detected. The results indicate the extremely low potential for off-target crossover to the major CNS drug target classes.

### 3.4. Effects of S-CE-123 on Neurite Outgrowth

To evaluate the cytotoxicity of S-CE-123, a neurite outgrowth was examined in rat primary cortical neurons. S-CE-123 (0.2–100E-06 M) was added to cultures of cortical neurons and cell counts, mean neurite average length and neurite total length per neuron were determined after 72 h. S-CE-123 did not have any significant effect on cell viability and neurite outgrowth (Figure 1D,E).

### 3.5. S-CE-123 Increased Locomotion/Exploration in Familiar, but Not in a Novel Environment 

The effect of S-CE-123 on spontaneous locomotor activity in naive male Sprague Dawley rats (10 weeks at the start of the experiment) was evaluated in the OF apparatus. In a novel OF test, there were no differences between vehicle and S-CE-123 at 10 mg/kg in distance travelled (*p* > 0.05, unpaired *t*-test, n = 7–8; Figure 2A).

In a familiar OF, S-CE-123 had a dose-dependent effect on total distance travelled over 10 min (F_1.848,16.63_ = 20.90; *p* < 0.0001, one-way ANOVA) as compared to vehicle (Figure 2D_1_D_2_). Dunnett´s post hoc test revealed that S-CE-123 at 10 and 20 mg/kg significantly increased distance travelled (*p* < 0.0001 and *p* = 0.0009 for 10 and 20 mg/kg, respectively) as compared to vehicle (0 mg/kg). The increase in exploration at 10 and 20 mg/kg was comparable to H1 (*p* > 0.05). Habituation sessions (H2 to H4) were interpolated to the study to control a possible cumulative effect of the compound (Figure 2E). Baseline activity measured during habituation sessions (H2-4) revealed no significant differences compared to vehicle (0 mg/kg) during test session (*p* > 0.05). The drug did not induce stereotypic or anxiogenic behavior (Figure 2B,C).

#### 3.5.1. A Hole-Board Test Did Not Show Differences between S-CE-123-Treated and Vehicle-Treated Rats at Young Age

Forty young male Sprague Dawley rats (5 months old) were randomly assigned to the treatment groups and were trained in reward-motivated spatial learning and memory task (Figure 3A). A two-way RM-ANOVA on RMIs and latency revealed a significant trial effect (F_5.099,183.6_ = 22.32, *p* < 0.0001; F_3.717,133.8_ = 15.15, *p* < 0.0001), but no treatment effect between the groups (F_3,36_ = 0.4642, *p* = 0.7090; F_3,36_ = 0.1431, *p* = 0.9335) and no trial × treatment interaction (F_27,324_ = 1.012, *p* = 0.4517; F_27,324_ = 0.8022, *p* = 0.7492) (Figure 3C,D). However, the finding of no differences between treatment groups in young animals may be due to a ceiling effect.

#### 3.5.2. Four Groups of Aged Rats Chosen for the Study Displayed Similar Ability for Spatial Learning and Memory 

A large cohort of male Sprague Dawley rats (22–24 months; n = 160) was characterized using a hole-board test [34]. Thirty-eight aged rats (22–23 months) used in the present study were chosen from “intermediate performers” (Figure 3E), which were scored based on their mean RMIs derived from trial 6 and 10 in the retrieval phases. Rats characterized as either inferior (having mean RMI < mean − 1SD) or superior (having mean RMI > mean + 1SD) were not applicable for the treatment and were excluded from this study [34]. The four groups of aged rats chosen for this study exhibited similar learning and memory ability prior to the commencement of treatment (Figure 3F). Two-way RM-ANOVA revealed progressive and comparable increases in RMIs over ten trials in all groups (F_3.256,110.8_ = 10.56, *p* < 0.0001). Hence, RMIs did not differ between the four groups (F_3,34_ = 0.4901, *p* = 0.6915).

#### 3.5.3. S-CE-123 Treatment Improves Ability for Spatial Learning and Memory Formation at Advanced Age

The effect of S-CE-123 (i.p., 0, 1, 5 and 10 mg/kg) on reward-motivated spatial learning and memory was evaluated using a hole-board test with characterized aged male Sprague Dawley rats (see paragraph above). Figure 3B illustrates the timeline of behavioral tests and treatment of aged rats. 

A two-way RM-ANOVA on RMIs on day1 revealed a significant trial effect (F_4,136_ = 14.32, *p* < 0.0001), trial × treatment interaction (F_12,136_ = 2.142, *p* = 0.018) and overall treatment effect (F_3,34_ = 4.050, *p* = 0.0145). Dunnett’s post hoc test revealed significantly increased RMIs in aged rats treated with 1 and 10 mg/kg compared to the vehicle group (*p* = 0.0334 and 0.0135, respectively). On day2, there was a significant treatment effect (two-way RM-ANONA, F_3,34_ = 8.021, *p* = 0.0004) and Dunnett’s post hoc test revealed significantly increased RMIs in treated groups 1 and 10 mg/kg compared to vehicle group (*p* = 0.0029 and <0.0002; respectively). One-way ANOVA revealed a significant treatment effect on day3 (F_3,34_ = 3.273, *p* = 0.0328); Dunnett’s post hoc test revealed significant effect in aged rats treated with 10 mg/kg compared to the vehicle group (*p* = 0.0406) (Figure 3G–I).

Old rats tend to be less active and spend more time immobile (Figure 3J). We compared immobility during the first trial on day1; aged S-CE-123-treated animals spent significantly less time immobile compared to the vehicle-treated group (*p* = 0.0038, Mann–Whitney test), whereas, in young animals, treatment did not have an effect on immobility (*p* = 0.4988, Mann–Whitney test) (Figure 3J).

Next, we compared the performance of aged rats in the first hole-board test (no treatment) and in the second test conducted at least 2 months later (treatment). Analyses of RMIs between the first and second test within four groups of aged rats showed that performance in a spatial memory task was improved in S-CE-123-treated groups at 1 and 10 mg/kg doses (Figure 4B–D), whereas in the vehicle group, performance neither improved nor worsened (Figure 4A). An RM-ANOVA on RMI showed an significant trial effect (F_4,32_ = 7.350, *p* = 0.0003 and F_4,36_ = 5.138, *p* = 0.002; 1 and 10 mg/kg, respectively) and treatment effect (F_1,8_ = 12.38, *p* = 0.0079 and F_1,9_ = 30.62, *p* = 0.0004; 1 and 10 mg/kg, respectively) on day1; a significant trial effect (F_3,24_ = 4.914, *p* = 0.0084 and F_3,27_ = 3.379, *p* = 0.0327; 1 and 10 mg/kg, respectively) and treatment effect (F_1,8_ = 11.93, *p* = 0.0086 and F_1,9_ = 9.210, *p* = 0.0141; 1 and 10 mg/kg, respectively) was found on day 2. Paired *t*-test on RMIs on day 3 revealed a significant treatment effect at both doses (*p* = 0.0322 and 0.0024, 1 and 10 mg/kg, respectively). When RMIs of vehicle-treated group were compared to performance in the first hole-board test, no significant treatment effect (F_1,9_ = 0.1125, *p* = 0.7450, day1; F_1,9_ = 1.075, *p* = 0.3270, day2; *p* = 0.3925, day3) was detected.

### 3.6. No Treatment Effect in Non-Rewarding Spatial Memory Task

An independent sample of naïve aged male Sprague Dawley rats (21–22 months) were trained in the MWM task (Figure 4E) in order to test whether the memory-enhancing effect of S-CE-123 at a dose of 10 mg/kg body weight was also observable in a no-reward-motivated spatial memory task. No differences in the acquisition (escape latencies; two-way ANOVA, F(1,18) = 1.245, *p* = 0.2793; n = 10) or retention (probe trial; time in target quadrant, unpaired *t*-test, *p* = 0.1288) of the MWM task could be determined (Figure 4F–I). There were also no differences in the average velocities or travelled distances between groups.

### 3.7. Acute Treatment with S-CE-123 Has No Effect on Neuroinflammation 

As spatial learning and memory tasks heavily depend on hippocampal function, we focused our further histological analyses on this region. As an indicator of neuroinflammation, we analyzed the number of CD68-positive cells. Although there is some CD68 expression in resting microglia [35], CD68 is associated with the phagosomal/lysosomal pathway of microglia and is therefore considered a marker of activated microglia [36,37]. Elevation of the CD68 was also repeatedly reported in aged subjects [38]. Moreover, it has been suggested that CD68 could play a role associated with cognitive decline in a subgroup of the normally aged population [39]. Aged rats had a significantly higher number of CD68-positive cells (Figure 5A,B) compared to the young animals, suggesting altered phagosomal/lysosomal processing of microglia in the old. Acute treatment with S-CE-123 had no effect on the number of CD68-positive cells in aged and young animals in the hippocampus (unpaired *t*-test, *p* > 0.05, n = 8–9/group). 

## 4. Discussion

In the present study, pharmacological properties of a highly selective DAT inhibitor, S-CE-123, were examined in a number of in vitro and in vivo assays to address its suitability as a probable novel therapeutic agent. S-CE-123 is an analogue of modafinil, which also binds to DAT with an atypical profile [28]. Like modafinil, S-CE-123 is a relatively weak (IC_50_ = 2.8E-06 M, [30]) but apparently very selective DAT inhibitor with respect to NET and SERT. Commonly used DAT inhibitors such as cocaine, amphetamine, methylphenidate and GBR12909 exert insufficient specificity across monoamine transporters, where, for example, action at NET causes cardiovascular adverse effects such as tachycardia, increased heart rate and blood pressure; vasoconstriction and hyperthermia are linked to serotonergic action [40]. S-CE-123 showed no agonist or antagonist activity against any of the 168 GPCRs tested in the functional cellular assay. Similarly, no positive hit was detected in an in vitro competition binding assay against 468 protein and lipid kinases representative of all major human kinase families. In addition, our compound does not promote neuronal apoptosis.

On the other hand, good bioavailability indicates that S-CE-123 at 10 mg/kg is able to reach the effective concentrations (above IC50), as supported by behavioral effects observed also at 1 mg/kg. Clearly, S-CE-123 at 10 mg/kg increases extracellular dopamine in the nucleus accumbens (NAc) and prefrontal cortex [31,33], both VTA innervated dopaminergic brain regions involved in learning and memory. Importantly, the increase in dopamine in NAc was limited to the core [31,33]. This finding suggests the different modes of action of S-CE-123 from psychostimulants, which preferentially activate the NAc shell DA [41,42,43]. The activation of NAc shell DA transmission rather than that in the core is essential for the reinforcing properties [44]. However, the reinforcing properties of S-CE-123 have not been addressed so far.

Dopamine drives exploratory behavior in novel environments [45]. Novel stimuli are known to excite dopamine neurons [46,47] and heighten signals in brain regions receiving dopaminergic input. During the first exposure of rats to the open field chamber, animals showed increased spontaneous locomotion/exploration as compared to the repeated exposure to the board. 

We examined the effect of S-CE-123 on spontaneous locomotion/exploration in a novel environment, and unlike cocaine [48,49], S-CE-123 did not induce stimulant effects on locomotion. However, in the familiar environment, the blockade of DAT by S-CE-123 mimicked a novelty-induced effect on spontaneous locomotion without induction of anxiogenic or stereotypic behavior. It is unclear if an S-CE-123-mediated increase in locomotion/exploration in a familiar environment is triggered by attentional orienting, enhanced motivation or increased motor output.

Aging is accompanied by profound changes in the dopamine system that negatively affect executive and cognitive functions. Cognitive flexibility is an executive function involving multiple cognitive processes that work together to adjust the thoughts and behaviors in response to changing demands of a situation without explicit instructions [50]. S-CE-123 has a positive effect on cognitive flexibility in young rats [30]; however, the effect of the compound on cognitive flexibility in aging animals has not been tested because aging rats are not able to form and shift attention.

Several lines of evidence indicate that DAT blockers have pro-motivational and pro-cognitive effects in animal models [51,52,53,54]. S-CE-123 has been shown to reverse the motivational impairments induced by tetrabenazine and increase selection of high-effort PROG lever pressing [31]. Importantly, this effect is not only measurable in young animals, but S-CE-123 has markedly enhanced motivation also in aged experienced rats [32]. Recent research has demonstrated that motivational incentives can modulate cognitive functions and enhance performance in different cognitive tasks [55].

The role of reward in driving responses in midbrain dopamine regions indicates that reward may also modulate interactions between midbrain dopamine regions, the PFC and the hippocampus [56,57]. Food reward and food-reward-associated stimuli elevate dopamine levels in the NAc and the PFC [58,59]. Despite cognitive deficits, the positive influence of reward anticipation on memory performance remains intact in old adults [60]. It is feasible that a DAT-inhibition-mediated increase in extracellular DA may support reward-motivated cognitive processes in low-stress-inducing tasks such as the hole-board test, whereas water maze exposure induces stress and elevated corticosterone release [61]. We used a reward-motivated spatial memory task, the hole-board test, to test the effect of S-CE-123 on learning and memory in young and aging animals. S-CE-123 was able to restore learning and memory functions in aged male Sprague Dawley rats in which cognitive performance was compromised as a result of the normal physiological aging processes. The test was performed on aged behaviorally characterized rats to reduce inhomogeneity of an aging population [34]. We observed a cognitive enhancing effect of S-CE-123 in old rats at 1 and 10 mg/kg. S-CE-123 shows a unique enhancing effect on memory acquisition after a single drug administration, a result consistent with the effects of the racemic form on memory acquisition tested on young compromised rats [28]. Moreover, significantly decreased latency in an S-CE-123 group may not only refer to enhanced learning and memory function but may also indicate increased motivation to perform a given task. DAT-inhibition-mediated enhancement of motivation [31,32] may play a particularly relevant role in modulation or even counteracting age-related changes in cognitive functions. 

The response to the compound is increased in old compared to young rats, which may be related to two reasons: first, cognitive processes in young rats may be less predominantly supported by dopaminergic processes but to a higher extent by other transmitters such as noradrenaline and thus adrenergic mechanisms. Second, differences in the dopaminergic system in aged as compared to young rats may support an increased benefit of elevated extracellular levels of dopamine. The first point is supported by the findings of Allard et al. [62] that spatially cognitively unimpaired aged rats show increased dopaminergic fibers in the neocortex rather than young or age-impaired rats, whereas no differences could be found for noradrenergic fibers. Further, age-related impairments in serotonergic or cholinergic functions [63] may be partially compensated by enhanced dopaminergic activity. 

In future studies, the effect of chronic treatment should be addressed. We cannot rule out that chronic DAT blockade may trigger adaptive changes [64] that can contribute to drug tolerance and both acute and chronic adverse effects. Neurotoxic effects derived from the oxidation of accumulated dopamine [65] cannot be ruled out either. However, much of the research on the parent compound modafinil has shown that the drug has long-term efficacy extending for as long as two years and is well tolerated, and no significant number of patients developed a drug tolerance [66,67].

## 5. Conclusions

Based upon the current studies reported herein, S-CE-123 may be expected to be a possible candidate for improving cognitive measures in patients with age-related cognitive decline as well as several psychiatric disorders with cognitive impairments.

## Figures and Tables

**Figure 1 biomolecules-13-00467-f001:**
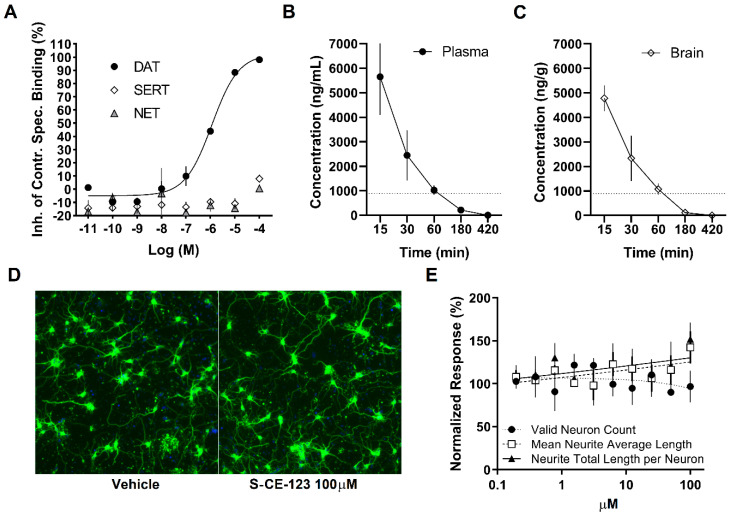
(**A**) S-CE-123 competes for the binding of radioligand specific to the hDAT with IC50 = 1.1E-06 M and K_i_ = 6.1E-07 M. (**B**) The plasma concentrations of S-CE-123 after a single i.p. administration at 10 mg/kg. (**C**) The brain concentrations of S-CE-123 after a single i.p. administration at 10 mg/kg. (**D**,**E**) S-CE-123 did not show neurotoxic effect on cell viability or neurite outgrowth of cortical neurons in vitro. Data are presented as a mean ± SEM, n = 3.

**Figure 2 biomolecules-13-00467-f002:**
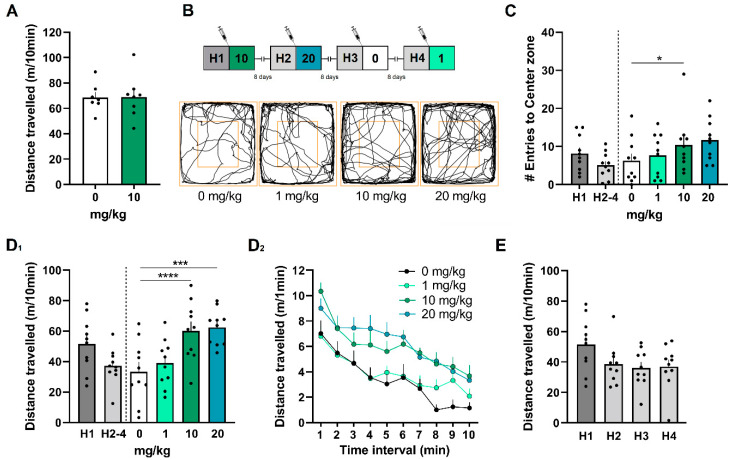
Effect of S-CE-123 on locomotor activity in an open field apparatus. (**A**) A novelty test did not show significant effects on distance travelled at 10 mg/kg (*p* > 0.05, unpaired *t*-test, n = 7–8). (**B**–**E**) Familiar OF. (**B**) Schematic demonstration of experimental design and representative traces of locomotor activity recorded during test sessions with 0, 1, 10 and 20 mg/kg S-CE-123. (**C**) Number of entries to the center zone. (**D_1_**) Total distance travelled over 10 min and (**D_2_**) 1 min intervals. (**E**) Baseline activity measured during habituation sessions (all rats received vehicle at 1 mL/kg; H1–novelty session, H2-4–familiar sessions). One-way RM-ANOVA followed by Dunnett post hoc test for multiple comparisons; * *p* < 0.05, *** *p* < 0.001, **** *p* < 0.0001. Values are presented as mean ± SEM, n = 10.

**Figure 3 biomolecules-13-00467-f003:**
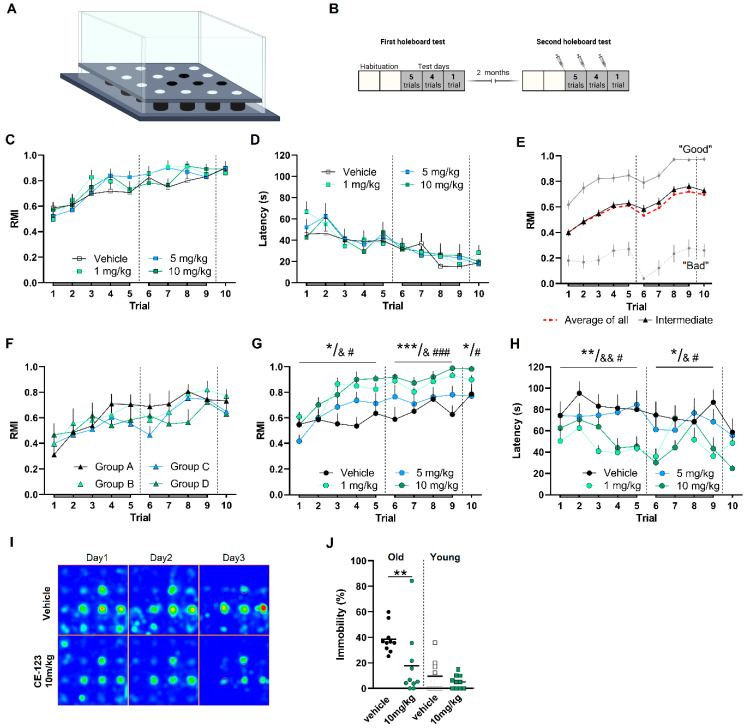
(**A**) Schematic of the hole-board maze, a non-sophisticated paradigm for the evaluation of reward-motivated spatial learning and memory. (**B**) A cartoon depicting the timeline of experiments. (**C,D**) Performance of young rats (5 months) treated with S-CE-123 at 1, 5 and 10 mg/kg body weight or vehicle in hole-board task. There were no significant differences in (**C**) RMIs (F_3,36_ = 0.4642, *p* = 0.7090, two-way RM-ANOVA) or (**D**) latency (F_3,36_ = 0.1431, *p* = 0.9335, two-way RM-ANOVA) between four groups (n = 10/group). (**E**) A large cohort of 22–24 months old rats (n = 160) [34] was behaviorally characterized using a hole-board test and based on their mean RMIs derived from trial 6 and 10, rats were characterized as impaired, unimpaired or “intermediate”. Thirty-eight “intermediate” rats of age 22–23 months were selected, randomly grouped and used for treatment. (**F**) Four groups of aged rats were equal in terms of RMIs in the first hole-board test. (**G**–**I**) Performance of old (24–25 months) rats treated with S-CE-123 at 1, 5 and 10 mg/kg body weight or vehicle in hole-board task (n = 9–10/group). There was a significant treatment effect with improved (**G**) RMIs (day1 (F_3,34_ = 4.050, *p* = 0.0145, two-way RM-ANOVA), day2 (F_3,34_ = 8.021, *p* = 0.0004, two-way RM-ANOVA); day3 (F_3,34_ = 3.273, *p* = 0.0328, one-way ANOVA) and decreased (**H**) latency (day1 (F_3,34_ = 5.405, *p* = 0.0038, two-way RM-ANOVA), day2 (F_3,34_ = 3.698, *p* = 0.0209, two-way RM-ANOVA); day3 (F_3,34_ = 2.281, *p* = 0.0969, one-way ANOVA) in S-CE-123 groups. * ANOVA general treatment effect, multiple comparison, & vehicle vs. 1 mg/kg and # vehicle vs. 10 mg/kg. (**I**) Heat map of hole visits of aged rats. (**J**) In the first trial on day1, aged S-CE-123-treated animals spent significantly less time immobile compared to vehicle treated rats (*p* = 0.0038, Mann–Whitney test), whereas, in young animals, treatment had no effect on immobility (*p* = 0.4988, Mann–Whitney test). Values are expressed as mean ± SEM, (*/&/# *p* < 0.05; **/&& *p* < 0.01; ***/### *p* < 0.001).

**Figure 4 biomolecules-13-00467-f004:**
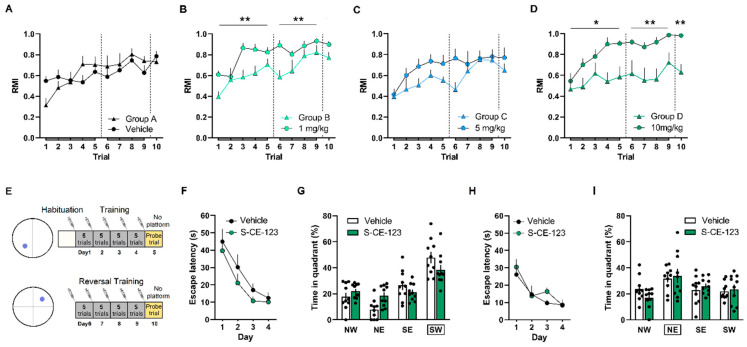
Analyses of hole-board RMIs in the first test (no treatment) compared to the second test (treatment) for (**A**) vehicle group (day1 (F_1,9_ = 0.1125, *p* = 0.7450, 2-way RM-ANOVA); day2 (F_1,9_ = 1.075, *p* = 0.3270, 2-way RM-ANOVA); day3 (*p* = 0.3925, paired *t*-test)) and for S-CE-123 groups at (**B**) 1 mg/kg (day1 (F_1,8_ = 12.38, *p* = 0.0079, 2-way RM-ANOVA); day2 (F_1,8_ = 11.93, *p* = 0.0086, 2-way RM-ANOVA); day3 (*p* = 0.0322, paired *t*-test)), (**C**) 5 mg/kg (day1 (F_1,8_ = 3.386, *p* = 0.1030, 2-way RM-ANOVA); day2 (F_1,8_ = 2.448, *p* = 0.1563, 2-way RM-ANOVA); day3 (*p* = 0.2071, paired *t*-test)) and (**D**) 10 mg/kg (day1 (F_1,9_ = 30.62, *p* = 0.0004, 2-way RM-ANOVA); day2 (F_1,9_ = 9.210, *p* = 0.0141, 2-way RM-ANOVA); day3 (*p* = 0.0024, paired *t*-test)). (**E**–**I**) Different sample of old rats (21–22 months) were tested in a MWM spatial memory task. (**E**) Schematic demonstration of experimental design and treatment. There was no significant difference between vehicle and 10 mg/kg S-CE-123 in (**F**) escape latency during four days of training, in (**G**) time spent in target quadrant during probe trial, (**H**) in escape latency during four days of training in reversal learning and in (**I**) time in target quadrant during probe trial of reversal test. Values are expressed as mean ± SEM (n = 10; * *p* < 0.05; ** *p* < 0.01).

**Figure 5 biomolecules-13-00467-f005:**
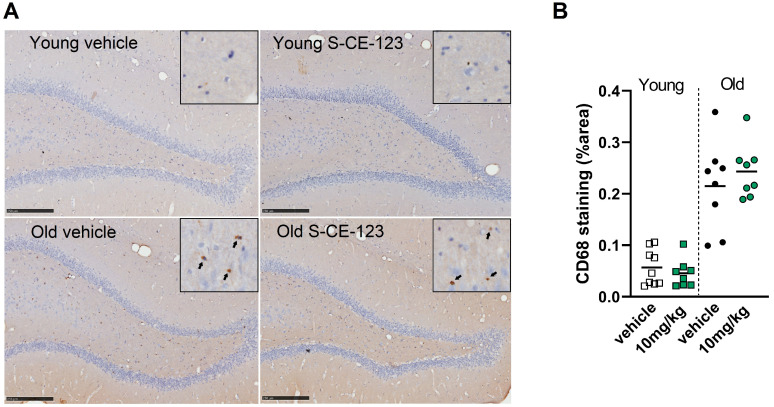
(**A**) Representative images of CD68 immunostaining in DG hippocampal subregion (scale bars represent 250 μm). (**B**) Quantification of CD68 immunostaining in the DG of young and aged rats treated with 10 mg/kg S-CE-123 or vehicle. Each dot represents the mean of values across the three sections in an individual animal. Values are expressed as mean ± SEM.

## Data Availability

The datasets generated during and/or analyzed during the current study are available from the corresponding author upon reasonable request.

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
