# Peer review of "Low-Affinity/High-Selectivity Dopamine Transport Inhibition Sufficient to Rescue Cognitive Functions in the Aging Rat"

_biomolecules, 2023, doi:10.3390/biom13030467_

Round 1
Reviewer 1 Report
In this manuscript, Lubec et al. describe an enantiomerically pure, highly specific DAT inhibitor, S-CE-123, synthesized in their lab. S-CE-123 was tested for its ability to enhance and/or rescue cognitive functions in young and aged rats in the non-invasive, reward-motivated paradigm of a hole-board test for spatial learning. They also conducted binding studies for DAT, NET, and SERT, GPCR and kinome screening, and pharmacokinetic and basic neurotoxic screening for S-CE-123. I do have some minor concerns regarding the manuscript.
- How was the dose of 10 mg/kg determined for pharmacokinetic studies? Is there any previous research on which the dose was based?
- For behavioral studies related to spatial learning, why is the 20 mg/kg dose not tested, whereas it has been used for locomotion tests?
- Is there any study being conducted to show the effect of chronic administration?
- How this compound is different from drugs of abuse like cocaine, amphetamine, etc. Can it be used to prevent drug addiction?
Author Response
- How was the dose of 10 mg/kg determined for pharmacokinetic studies? Is there any previous research on which the dose was based?
We selected concentration based on the study with the racemic form (Cit. 28 - Kristofova, 2018). Because we could not run the analysis, we had to contract the company. However, our resources are limited.
- For behavioral studies related to spatial learning, why is the 20 mg/kg dose not tested, whereas it has been used for locomotion tests?
In a spatial learning task we could see the effect already at lower doses, what was expected also based upon the study on young compromised rats with the racemate. Also the goal was to access the effect in aged cognitively compromised rats, where the number of available rats was limited and animals could not be re-tested with different doses.
Is there any study being conducted to show the effect of chronic administration?
The longest treatment was 10 days at 10mg/kg, no adverse effects were observed.
- How this compound is different from drugs of abuse like cocaine, amphetamine, etc. Can it be used to prevent drug addiction?
Cocaine and amphetamine are not specific to DAT but target also SERT and NET. S-CE-123 is not releasing DA compared to amphetamine and has a distinct DAT binding mode from that shown by cocaine and cocaine-like compounds (Cit. 28 - Kristofova, 2018).
We can only speculate about the possible use of S-CE-123 to treat/ prevent drug addiction. However, studies on the parental compound modafinil indicated limited abuse potential but also limited potential for treating drug addiction.
Reviewer 2 Report
This is an extremely well written presentation of an interesting experiment evaluating the effects of a novel dopamine transporter inhibitor on cognitive and non-cognitive behavioral measures. The methods are sound, results clearly presented, and the interpretations appropriate. I have only a few minor comments.
Line 57: 'has', not 'have'
Line 66: delete 'a'
Lines 85-86: The authors speak about off-target behavioral actions of dopamine transporter inhibitors that also have actions on NET and SERT, but do not provide any specific examples. Providing examples, will help justify the need for a specific inhibitor.
Line 191: Could the rats see the platform in the Morris water maze?
Line 226: glass slides (not glasses)
Lines 494-513: There is an issue with the font change.
Tangential question: The drug is intended to treat cognitive issues associated with aging. I was curious if the drug might have abuse potential.
Author Response
Line 57: 'has', not 'have'
Was corrected.
Line 66: delete 'a'
Was corrected.
Lines 85-86: The authors speak about off-target behavioral actions of dopamine transporter inhibitors that also have actions on NET and SERT, but do not provide any specific examples. Providing examples, will help justify the need for a specific inhibitor.
Examples are provided in the first paragraph of the discussion.
Line 191: Could the rats see the platform in the Morris water maze?
Water tank was black same as platform placed below the water surface; animal couldn’t see the platform.
Line 226: glass slides (not glasses)
Was corrected.
Lines 494-513: There is an issue with the font change.
Was corrected.
Tangential question: The drug is intended to treat cognitive issues associated with aging. I was curious if the drug might have abuse potential.
The abuse potential of CE-123 remains to be addressed. There is only indirect evidence based on the fact that CE-123 lacks the effect on 50kHz USV, which is an emission marker for compounds with abuse potential (Citation 33, Sagheddu, 2020).
Reviewer 3 Report
The study by Dr. Lubec et al. focuses on the evaluation of the potency of low affinity DAT inhibitor in mitigating the cognitive decline in aging. Indeed, in the course of physiological aging, there is a progressive decrease in panneuronal dopaminergic innervation, which many researchers attribute to age-associated cognitive decline. The article is well-written and accessible, and the findings will be of interest to a wide range of readers. However, the material and method section lacks important piece of information, and the discussion section needs more objectivity. Relying on this, I recommend accepting the manuscript after major revision.
Major claim
1. In the "Materials and Methods" section, the subsection describing the animals (origin, line, housing conditions, age, formation of representative groups, volume of samples) is completely missing. We found some information in the captions to the figures, but this does not completely negate the need to write a full and detailed subsection "Animals" in the "Materials and Methods" section.
2. The authors avoid discussing the limitation of the studies, namely the possible negative effects of selective DAT blockade: 1) Adaptive changes in the dopaminergic system, particularly decreased dopamine receptor expression levels, indicating time-limited effects of the proposed therapy (https://doi.org/10.1016/S0165-0173(97)00063-5 ; https://doi.org/10.1046/j.1460-9568.1999.00764.x, for example) and 2) Toxic effects of extracellular dopamine, possible, through its oxidation (for example, https://doi.org/10.1002/syn.20554), which may lead to delayed side effects. These points should be discussed in the appropriate section of the manuscript.
Author Response
- In the "Materials and Methods" section, the subsection describing the animals (origin, line, housing conditions, age, formation of representative groups, volume of samples) is completely missing. We found some information in the captions to the figures, but this does not completely negate the need to write a full and detailed subsection "Animals" in the "Materials and Methods" section.
We added a general section, Animals and animal ethics, but as stated in this section, details on age and numbers were kept in specific sections to make it easier to follow. All missing informations were added.
- The authors avoid discussing the limitation of the studies, namely the possible negative effects of selective DAT blockade: 1) Adaptive changes in the dopaminergic system, particularly decreased dopamine receptor expression levels, indicating time-limited effects of the proposed therapy (https://doi.org/1016/S0165-0173(97)00063-5 ; https://doi.org/10.1046/j.1460-9568.1999.00764.x, for example) and 2) Toxic effects of extracellular dopamine, possible, through its oxidation (for example, https://doi.org/10.1002/syn.20554), which may lead to delayed side effects. These points should be discussed in the appropriate section of the manuscript.
We tried to address these points in the discussion.
Reviewer 4 Report
Thanks to the authors of this article, which is well done and has no significant deficiency.
My question is why S-CE-123 did not increase locomotion in the novel environment? Doesn't this indicate anxiety-like behavior?
If this drug increases motivation, why are the rats not motivated to explore in the novel environment?
Can the role of serotonin or norepinephrine be important in this regard?
Considering the half-life of the drug, how does it affect cognitive tests that last several days? MWM test was done in several days, which was not significant. Could it (half-life) be related to intact spatial memory?
Also, considering S-CE-123 half-life, can it be true that the interval between the injection of this drug and the memory test (perhaps several days later) could affect the results? Or is the effect of S-CE-123 on acquisition/retrieval different?
Of course, in the MWM, I have not noticed the number and timeline on injections.
The timeline and schedule of drug injection are not clearly explained; Exactly how and at what times the drug was injected in each test. Please add this item for each test in the method section.
Author Response
My question is why S-CE-123 did not increase locomotion in the novel environment? Doesn't this indicate anxiety-like behavior?
Introduction of animal to novel environment induces DA release and increased exploration/locomotion. This was not altered by S-CE-123, neither increased nor decreased. There was no difference in the center crossing, so no anxiogenic effect of the compound was detected.
If this drug increases motivation, why are the rats not motivated to explore in the novel environment?
S-CE-123-treated rats were motivated to explore novel environment in the same extent as control rats.
Can the role of serotonin or norepinephrine be important in this regard?
At doses studies, S-CE-123 has no or negligible effect on NET and SERT, so direct modulation of NE and SER system is unlikely.
Considering the half-life of the drug, how does it affect cognitive tests that last several days? MWM test was done in several days, which was not significant. Could it (half-life) be related to intact spatial memory?
The drug was administered every day 30 min prior to the behavioural test. Of course, the duration of the treatment, as well as timing, may have an effect on performance. We can only conclude that described treatment protocol didn’t have positive or negative impact on performance in MWM.
Also, considering S-CE-123 half-life, can it be true that the interval between the injection of this drug and the memory test (perhaps several days later) could affect the results? Or is the effect of S-CE-123 on acquisition/retrieval different?
The timing of the treatment has a sure impact on the results. Especially for aged animals treatment has a pronounced effect on the acquisition; clearly, they learned better already after a single injection. It would be interesting to see if it would be enough to measure differences also in memory retrieval; however, we didn’t apply different treatment protocol, only the one described.
Of course, in the MWM, I have not noticed the number and timeline on injections.
This information was added to methods and also to the Figure 4
The timeline and schedule of drug injection are not clearly explained; Exactly how and at what times the drug was injected in each test. Please add this item for each test in the method section.
This Information was added to the method section.
Round 2
Reviewer 3 Report
The authors did a good job on the manuscript. I recommend accepting the article in its current form
Reviewer 4 Report
Thanks to the authors for their effort
They have addressed my previous comments
In my opinion, the revised version is acceptable